# TGFβ-dependent expression of PD-1 and PD-L1 controls CD8[+] T cell anergy in transplant tolerance

Marije Baas[1,2,3†‡], Alix Besançon[1,2,3†], Tania Goncalves[1,2,3], Fabrice Valette[1,2,3], Hideo Yagita[4], Birgit Sawitzki[5], Hans-Dieter Volk[5,6], Emmanuelle Waeckel-Enée[1,2,3], Benedita Rocha[1,7], Lucienne Chatenoud[1,2,3], Sylvaine You[1,2,3*]

[1]University Paris Descartes, Sorbonne Paris Cité, Paris, France; [2]Institut National de la Santé et de la Recherche Médicale Unit 1151, Institut Necker-Enfants Malades, Paris, France; [3]Centre National de la Recherche Scientifique UMR 8253, Institut Necker-Enfants Malades, Paris, France; [4]Department of Immunology, Juntendo University School of Medicine, Tokyo, Japan; [5]Institute of Medical Immunology, Charité University Medicine, Berlin, Germany; [6]Berlin-Brandenburg Center for Regenerative Therapies, Charité University Medicine, Berlin, Germany; [7]Lymphocyte Population Biology Unit, Pasteur Institute, Paris, France

*For correspondence: sylvaine. you@inserm.fr

[†]These authors contributed equally to this work

Present address: [‡]Department of Nephrology, Radboud University Medical center, Nijmegen, The Netherlands

Competing interests: The authors declare that no competing interests exist.

**Abstract** CD8[+] T cell anergy is a critical mechanism of peripheral tolerance, poorly investigated in response to immunotherapy. Here, using a pancreatic islet allograft model and CD3 antibody therapy, we showed, by single cell gene profiling, that intragraft CD8[+] lymphocytes coexpressing granzyme B and perforin were selectively depleted through the Fas/FasL pathway. This step led to long-standing anergy of the remaining CD8[+] T cells marked by the absence of cytotoxic/ inflammatory gene expression also confirmed by transcriptome analysis. This sustained unresponsiveness required the presence of the alloantigens. Furthermore, tissue-resident CD8[+] lymphocytes produced TGFβ and expressed the inhibitory receptors PD-1 and PD-L1. Blockade of TGFβ downregulated PD-1 and PD-L1 expression and precipitated graft rejection. Neutralizing PD-1, PD-L1 or TGFβRII signaling in T cells also abrogated CD3 antibody-induced tolerance. These studies unravel novel mechanisms underlying CD8[+] T cell anergy and reveal a cell intrinsic regulatory link between the TGFβ and the PD-1/PD-L1 pathways.

## Introduction

Upon antigen recognition, CD8[+] T lymphocytes proliferate vigorously and differentiate into effector cells characterized by their ability to produce cytokines and chemokines, to migrate to inflamed tissues and to use various cytolytic pathways to kill their targets (*Williams and Bevan, 2007*). CD8[+] T cells play major role in transplant rejection. Through direct presentation, they get activated by donor antigen-presenting cells (APC), in particular dendritic cells, which migrate from the graft to secondary lymphoid organs (*Gras et al., 2011*; *Ochando et al., 2006*). This response is rapid, intense and induces acute graft rejection. Recipient APC also capture donor antigens from the transplant, present them to alloreactive CD8[+] T cells through self MHC I molecules (cross-priming/indirect presentation) hereby contributing to acute and chronic graft rejection (*Celli et al., 2011*; *Valujskikh et al., 2002*). Cytotoxic CD8[+] T lymphocytes migrate to the graft where they destroy target cells through granzyme B/perforin- and/or Fas/FasLigand-dependent cytolytic pathways.

**eLife digest** The immune system is always on guard for signs of infection or cells that have become diseased. When these signs are identified, a subset of white blood cells called CD8$^+$ T cells leap into action, multiply in number and then act to eliminate the potential threat. While this response is essential to fighting off infections and other diseases like cancer, it can backfire in people with an organ transplant. Indeed, the CD8$^+$ T cells can target and attack the cells of the transplanted organ causing the body to reject the organ.

One way to avoid transplant rejection would be to turn off CD8$^+$ T cells that have learned to recognize cells from the transplant. In fact, studies in 2012 and 2013 showed that treating transplanted animals with an antibody that binds T cells protects a transplanted organ from attack. This treatment had to be given after the CD8$^+$ T cells had recognized and began targeting the transplanted organ to be effective. But it was not clear exactly how this antibody treatment protected the transplant.

Now, Baas, Besançon et al. – including some of the same researchers involved in the earlier studies – show that the antibodies used in the treatment selectively target and eliminate the attacking CD8$^+$ T cells. This leaves behind only inactive CD8$^+$ T cells that don't harm the transplant. To do this, Baas, Besançon et al. transplanted pancreatic cells from mice into other mice with a diabetes-like disorder. Next, the experiments compared gene expression in CD8$^+$ T cells found within the transplanted tissue in mice that were treated with the antibody and those that were not treated. The expression of many genes for toxic molecules was stopped after treatment with the antibody leaving the CD8$^+$ T cells in an inactive state.

In addition, the treated CD8$^+$ T cells expressed more of a certain type of receptor (called PD-1 and PD-L1) that acts as inhibitory checkpoint for the immune system. So, Baas, Besançon et al. treated transplanted mice with both the T cell-eliminating antibody and antibodies that block these inhibitory receptors to see what would happen. The transplanted organs were quickly attacked and rejected. This shows that the inhibitory receptors play a crucial role in helping to shut down attacking CD8$^+$ T cells in the initial antibody treatment and allowed long-term survival of the transplanted organs. Blocking another protein called TGFβ in antibody-treated mice also caused organ rejection. The findings help explain how these antibodies protect transplanted organs and may help scientists trying to develop new anti-transplant rejection drugs in the future.

In most tolerance promoting protocols, long-term graft survival was associated with CD8$^+$ T cell dysfunction which mainly resulted from clonal deletion and/or anergy (*Iwakoshi et al., 2000*; *Monk et al., 2003*; *Qian et al., 1997*). Classically defined as the functional inactivation of T cells to cognate antigens, anergy was first described *in vitro* when T cells recognized antigens (signal 1) in absence of appropriate costimulation (signal 2), usually provided by CD28 (*Schwartz, 2003*). T cells were not able to produce IL-2, entered a hyporesponsive non proliferative state that prevented further responses upon antigen re-encounter. Over the last decade, better insight was gained into the signaling events leading to anergy, highlighting in particular the role of the transcription factors NF-AT (nuclear factor of activated T cells) and early growth response gene 2 and 3 (Egr-2, Egr-3) (*Macian et al., 2002*; *Safford et al., 2005*). However, characterization of the anergic phenotype and gene signature as well as the mechanisms that drive and sustain CD8 T cell anergy *in vivo* in the transplant setting are incompletely understood.

Recently, we reported that CD3 antibodies (CD3 Abs) induced tolerance in fully MHC-mismatched experimental models of pancreatic islet and cardiac transplantation (*Goto et al., 2013*; *You et al., 2012*). The timing of treatment was crucial; long-term graft acceptance was obtained only when CD3 Abs were administered once T cell priming had occurred, a few days post-transplant. Permanent survival of a second graft from the original donor but not from third-party donor demonstrated that antigen-specific tolerance had been induced. Purified spleen CD8$^+$ T cells from CD3 Ab-treated recipients were unable to respond when stimulated with donor antigens while response to third-party antigens was conserved (*You et al., 2012*). In this model, CD3 Abs preferentially targeted and depleted activated effector T cells, notably within the graft.

In the present manuscript, we addressed the molecular mechanisms underlying CD8[+] T cell tolerance induced by CD3 Ab therapy. We conducted detailed analysis of alloreactive CD8[+] T cells combining single-cell gene profiling, transcriptome analysis and *ex vivo/in vivo* functional studies. We found that CD3 Abs selectively deleted *Gzmb*[+]*Prf1*[+] CD8[+] cytotoxic effectors within the transplant. CD8[+] T cells escaping this deletion became anergic. The presence of the alloantigen was mandatory for the effect just as was TGFβ signaling to promote and sustain PD-1/PD-L1-mediated CD8[+] T cell tolerance.

## Results

### CD3 Ab therapy selectively depletes *Gzmb*[+]*Prf1*[+] CD8[+] T cells and promotes anergy

We previously showed that CD3 Ab-induced transplant tolerance was associated with a drastic reduction of CD8[+] T cell infiltrates and of peripheral donor-specific CD8[+] T cell responses (*You et al., 2012*). Here we measured the anti-donor reactivity of graft infiltrating T cells using a 20 hr-IFNγ Elispot assay. Pancreatic islets from BALB/c mice were isolated and grafted under the kidney capsule of diabetic C57BL/6 recipients. Tolerogenic treatment with CD3 Ab F(ab')$_2$ fragments was applied for 5 days (50 µg/day) at day 7 after transplantation. Intragraft T cells recovered after CD3 Ab treatment, on days 14 or 100 post-transplant, did not respond to BALB/c donor antigens as opposed to graft infiltrating T cells of untreated recipients analyzed few days before rejection (day 14) (*Figure 1—figure supplement 1*).

To better dissect the impact of CD3 Ab therapy on alloreactive CD8[+] T lymphocytes, we took advantage of a validated multiplex single cell PCR method established by the group of B. Rocha. This technique provides information on cell heterogeneity through the analysis of the simultaneous expression of selected inflammatory and/or cytotoxic genes by individual CD8[+] T cells (*Peixoto et al., 2007*). We focused our analysis on Th1 and cytotoxic genes as it has been shown that the IFNγ, perforin and Fas/FasL pathways constituted predominant mechanisms of CD8[+] T cell-mediated destruction of islet allografts (*Diamond and Gill, 2000*; *Sleater et al., 2007*). Individual CD8[+] T cells were sorted from the islet allografts (72 cells) or spleen (48 cells) recovered from 3 individual recipients on day +14, that is right after the last injection of CD3 Abs, or on day +100 post-transplant, once tolerance was established. On day 14 post-transplant, in untreated recipients, graft infiltrating CD8[+] T cells expressed the cytolytic molecules *Gzmb*, *Prf1* and *Fasl* as well as *Tbx21* and *Klrg1* (*Figure 1A*). Thirty three percent of these cells co-expressed 3 or more of the 7 genes tested (*Figure 1B*). Interestingly, *Gzmb* was co-expressed with either *Prf1* or *Fasl* which rarely overlapped, suggesting the presence of two distinct subsets of graft infiltrating CD8[+] lymphocytes (*Figure 1C*). *Tbx21*, *Eomes* and *Klrg1* were preferentially associated with *Prf1* rather than *Fasl* (*Figure 1C*).

In CD3 Ab-treated recipients, on day +14 after transplantation, expression of *Prf1*, *Tbx21*, *Klrg1* and *Gzmb* by intragraft CD8[+] T cells was clearly reduced as compared to untreated mice (*Figure 1A*). The frequency of cells coexpressing 3 or more genes was significantly decreased (from 33.3% to 15.3%) while the number of cells expressing only one gene doubled after CD3 Ab treatment (*Figure 1B*). A dramatic decrease in *Gzmb*[+]*Prf1*[+] CD8[+] T cells was observed (*Figure 1C*). Contrasting with these findings, *Fasl* expression was enhanced as compared to untreated controls (50% versus 30.6%). The increased FasL expression on intragraft CD8[+] T cells was confirmed by flow cytometry (*Figure 1D*). Among this *Fasl*[+] subset, 38% co-expressed *Gzmb* (versus 62% in untreated recipients) and 33.3% did not express any other effector genes.

On day 100 post-transplant, expression of *Fasl*, *Prf1*, *Tbx21* and *Eomes* was detected in less than 5% of intragraft CD8[+] T cells in CD3 Ab-treated tolerant mice (*Figure 1A*). Only 5.6% of CD8[+] T cells coexpressed 3 or more genes and 43.1% of the cells did not express any of the 7 inflammatory/cytotoxic genes tested (*Figure 1B,C*). *Gzmb* was detected in 26% of the cells but was not associated with *Fasl* or *Prf1* suggesting the absence of potent cytotoxic function (*Figure 1A–C*).

In the spleen, the single cell gene profile differed from that of intragraft T cells. A large proportion (56%) of CD8[+] T cells from untreated mice coexpressed *Gzmb*, *Tbx21* and *Klrg1* but nor *Prf1* or *Fasl* on day +14 post-transplant (*Figure 1—figure supplement 2*). Such coexpression was abrogated after CD3 Ab therapy as 52% and 37.5% of spleen CD8[+] T cells expressed none or only one of the 7 selected genes, respectively (*Figure 1—figure supplement 2B and 2C*).

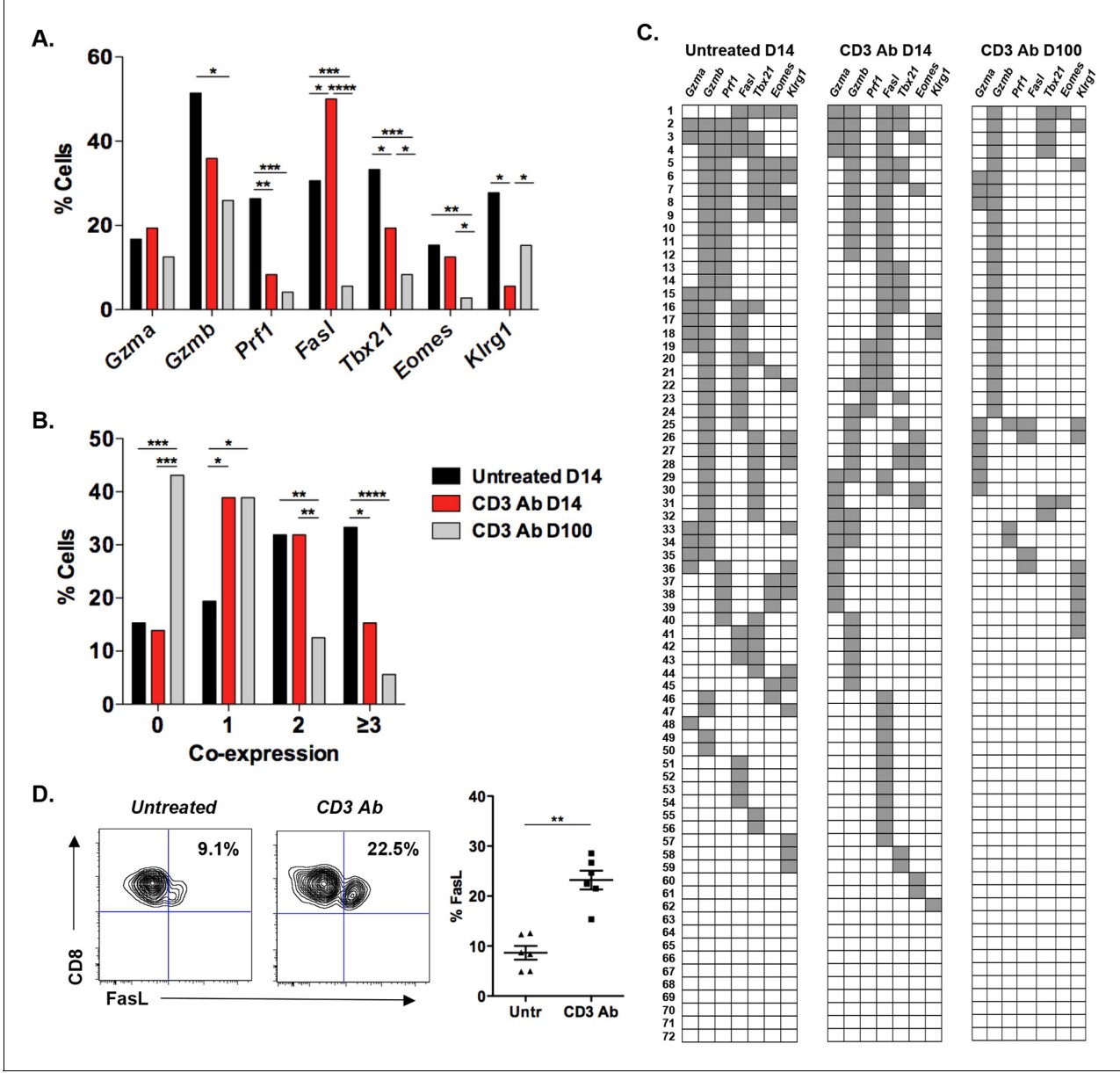

**Figure 1.** Coexpression of effector genes in graft-infiltrating CD8+ T cells after CD3 antibody therapy. C57BL/6 mice were transplanted under the kidney capsule with BALB/c pancreatic islets and treated or not with CD3 Abs on day +7 post-transplant. Individual CD8+ T cells (n = 72) present within the graft were FACS sorted on day +14 or day +100 post-transplant and subjected to multiplex gene expression analysis. (A) Proportion of CD8+ T cells among the 72 cells tested that expressed *Gzma*, *Gzmb*, *Prf1*, *Fasl*, *Tbx21*, *Eomes* and *Klrg1* mRNA in each group. (B) Polyfunctionality distribution of intragraft CD8+ T cells. (C) Coexpression of inflammatory and cytotoxic molecules by individual graft infiltrating CD8+ T cells. Each row represents one individual cell that is numbered. Each column represents a different gene. For better visualization of coexpression patterns, individual cells were ordered by the degree of gene coexpression. (D) FasL expression by intragraft CD8+ T cells 14 days after transplant and CD3 Ab therapy. (*Figures 1A and 1B*: $\chi^2$ test, *p<0.03, **p<0.01, ***p<0.0003, ****p<0.0001).

The following figure supplements are available for figure 1:

**Figure supplement 1.** IFNγ responses by intragraft T cells after CD3 Ab therapy.

**Figure supplement 2.** Multiplex gene expression analysis on individual splenic CD8+ T cells after CD3 Ab therapy.

**Figure supplement 3.** Expression of inflammatory, cytotoxic and apoptotic markers by CD8+ T cells.

RT-qPCR on total islet allografts confirmed the decreased expression of *Gzmb*, *Prf1*, *Tbx21* and *Ifng* (*Figure 1—figure supplement 3A*). *Fasl* expression was also reduced on day +14 post-transplant which contrasts with results from single cell PCR (*Figure 1*). We also analyzed expression of *Fas* mRNA which, interestingly, increased after administration of CD3 Abs on day +14 post-transplant and decreased hereafter. We confirmed by flow cytometry that CD3 Abs rapidly induced Fas expression on CD8$^+$ T cells after a 24 hr *in vitro* stimulation and most of them co-expressed FasL at 48 hr (*Figure 1—figure supplement 3B*).

To investigate whether CD3 Ab-induced depletion of alloreactive CD8$^+$ T cells was dependent on the Fas/FasL pathway, monoclonal antibodies neutralizing FasL were administered in C57BL/6 recipients at the time of CD3 Ab therapy. Long-term islet allograft survival was not observed in this condition (*Figure 2A*). FasL blockade abrogated CD3 Ab depleting effect: CD8$^+$ T cells were detected within the islet allografts at levels comparable to that found in untreated recipients (*Figure 2B*). FasL expression was drastically reduced on infiltrating CD8$^+$ T cells (*Figure 2C*).

## Graft infiltrating CD8$^+$ T cells present an inhibitory phenotype after CD3 Ab therapy

We next characterized intragraft CD8$^+$ T cells not eliminated by CD3 Ab treatment. On day 14 post-transplant, most of them were CD69$^+$ CD44$^{high}$CD62L$^{low}$CD45RB$^{low}$ (*Figure 3A* and *Figure 3—figure supplement 1*). They expressed low levels of CD122 (the β subunit of the IL-2 and IL-15 receptors) and CD25 as opposed to intragraft CD8$^+$ T cells from untreated mice (*Figure 3A*). Expression of the proliferation marker Ki67 was also strongly reduced after CD3 Ab therapy. In contrast, the inhibitory receptors PD-1, PD-L1 and LAG3 were upregulated (58.5%, 60.9% and 58.6% versus 25%, 26.1% and 27% without treatment, respectively) and were mostly co-expressed by the same cells (*Figure 3A,B*). This phenotype was maintained over long-term since a large proportion of intragraft CD8$^+$ T cells still expressed these inhibitory receptors on day +100 post-transplant. A similar trend was observed in the spleen and draining lymph nodes where CD8$^+$ T cells showed an increase in PD-1, PD-L1 and LAG-3 expression after CD3 Ab therapy (*Figure 3—figure supplement 2*).

Interestingly, CD3 antibody-induced upregulation of PD-L1 on intragraft CD8$^+$ T cells was inhibited following *in vivo* FasL blockade (*Figure 3C*). In addition, expression of CD25 and T-bet were significantly increased in recipients treated with the combination of CD3 and FasL Abs as compared to CD3 Abs alone, reaching levels similar to those observed in untreated mice (*Figure 3C*).

## Continuous presence of alloantigen and triggering of the PD-1/PD-L1 pathway are mandatory for the induction and maintenance of CD3 Ab-mediated CD8$^+$ T cell tolerance

To investigate the role of the PD-1/PDL-1 pathway in CD3 Ab-induced transplant tolerance, we treated C57BL/6 mice at day 7 with CD3 Abs together with monoclonal Abs to PD-1 or PD-L1. In both cases, recipients rejected their graft with a median survival of 33.7 ± 1.5 days or 28.6 ± 5.1 days, respectively (*Figure 4A,B*). After administration of PD-1 Abs, most graft infiltrating CD8$^+$ T cells were CD44$^{high}$CD62L$^{low}$Tbet$^+$ and strongly proliferated (*Figure 4—figure supplement 1A*). Anti-PD-L1 Abs were also injected in tolerant recipients, on day 100 after transplantation and CD3 Ab therapy. Blockade of PD-L1 at the time of established tolerance precipitated graft rejection (*Figure 4C*).

To further dissect the role of this pathway in CD8$^+$ T cell tolerance, we adoptively transferred purified spleen CD8$^+$ T cells from CD3 Ab-treated tolerant mice into immunodeficient RAG$^{-/-}$ recipients transplanted on the same day with BALB/c pancreatic islets. Islet grafts survived in these recipients. Transferred CD8$^+$ T cells were detected in significant proportion in the spleen, draining lymph nodes and within recipients' transplants (*Figure 4D–E*); they stained CD44$^{high}$ but did not produce high amounts of IFNγ in response to PMA/ionomycin stimulation (*Figure 4E* and *Figure 4—figure supplement 1B*). Administration of PD-L1 Abs on day +100 post-transplant provoked graft rejection (*Figure 4C*) and restored the capacity of CD8$^+$ T cells to secrete IFNγ (*Figure 4D–E* and *Figure 4—figure supplement 1B*).

To assess the role of donor alloantigens in sustaining CD8$^+$ T cell unresponsiveness, RAG$^{-/-}$ C57BL/6 recipients were grafted under the kidney capsule with BALB/c pancreatic islets either on the same day or more than two months after transfer of purified CD8$^+$ T cells recovered from

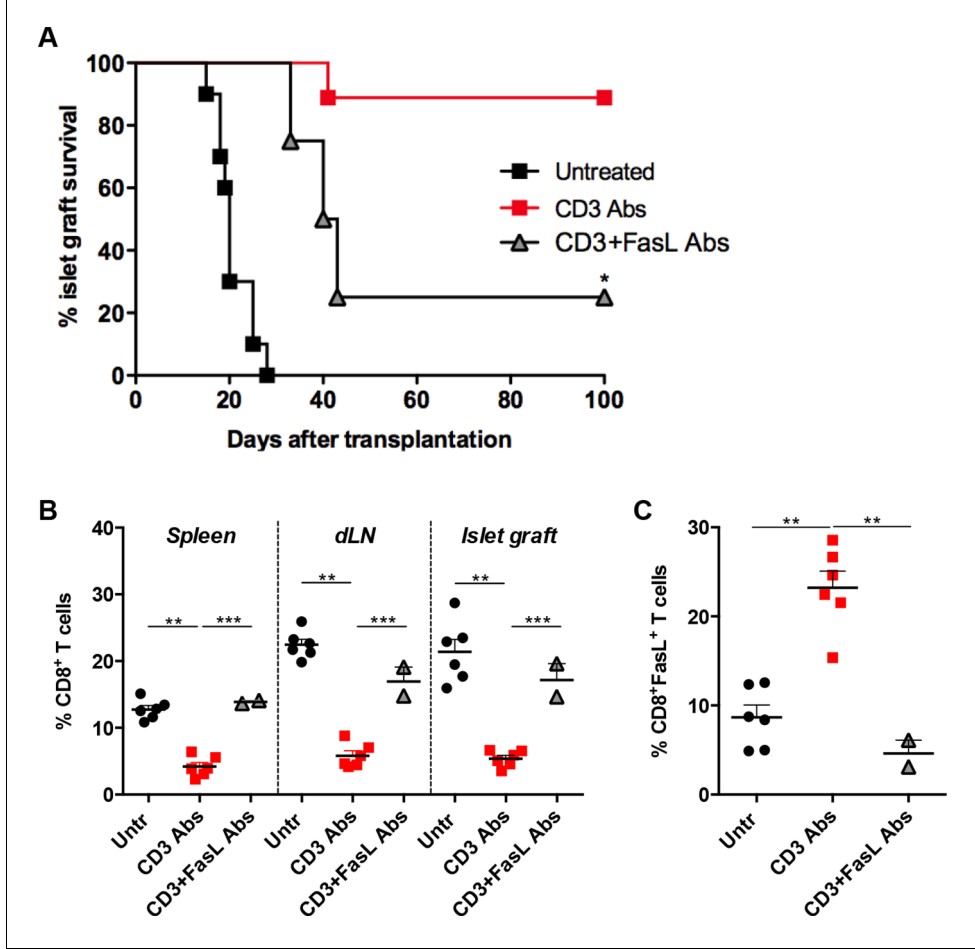

**Figure 2.** FasL blockade reversed CD3 antibody-induced transplant tolerance. (**A**) Graft survival of BALB/c islets was measured in C57BL/6 mice treated at day 7 with CD3 antibodies (50 µg, 5 days) alone (n = 10) or combined with neutralizing antibodies against FasLigand injected at the dose of 500 µg i.p. on day 6, 7 and 8 post-transplant (n = 4) (*p<0.02 between anti-CD3 and anti-CD3+anti-FasL Ab-treated mice). (**B**) Additional untreated (n = 6), CD3 Ab (n = 6) or CD3+FasL Ab (n = 2)-treated mice were sacrificed on day +14 post-transplant and proportion of CD8$^+$ T cells was analyzed in the spleen, renal draining lymph nodes (dLN) and the islet allografts (**p<0.003, ***p<0.0006). (**C**) Expression of FasL by graft infiltrating CD8$^+$ T cells isolated on day +14 from untreated (n = 6), CD3 Abs (n = 6) or CD3+FasL Abs (n = 2)-treated recipients (**p<0.002).

tolerant mice, to exclude any impact of homeostatic proliferation on T cell functional properties. In this later condition, all grafts were rejected contrasting with the long-term survival obtained when cell transfer and islet grafts were both performed on day 0. (*Figure 5*). A 4-week interval between infusion of tolerant CD8$^+$ T cells and islet allografts also led to graft rejection (*Figure 5*). Control syngeneic islet grafts were tolerated.

## Efficient TGFβ/TGFβR signaling promotes PD-1 and PD-L1 expression on graft infiltrating CD8$^+$ T cells and is required for CD3 Ab-induced transplant tolerance

We next analyzed the role of TGFβ in our model. RT-qPCR results on islet grafts recovered from CD3 Ab-treated recipients showed a significant increase of *Tgfb1* mRNA, as compared to untreated controls (*Figure 6A*). In addition, the intragraft *Tgfb1/Ifng* ratio was increased over long-term. To further identify the source of TGFβ, we sorted individual T lymphocytes, recovered on day 14 post-transplant, either from the spleen or from the islet grafts of 3 recipient mice treated with CD3 Abs. Expression of *Tgfb1* mRNA was determined by single cell qPCR on 60 CD8$^+$ and 48 CD4$^+$ T cells.

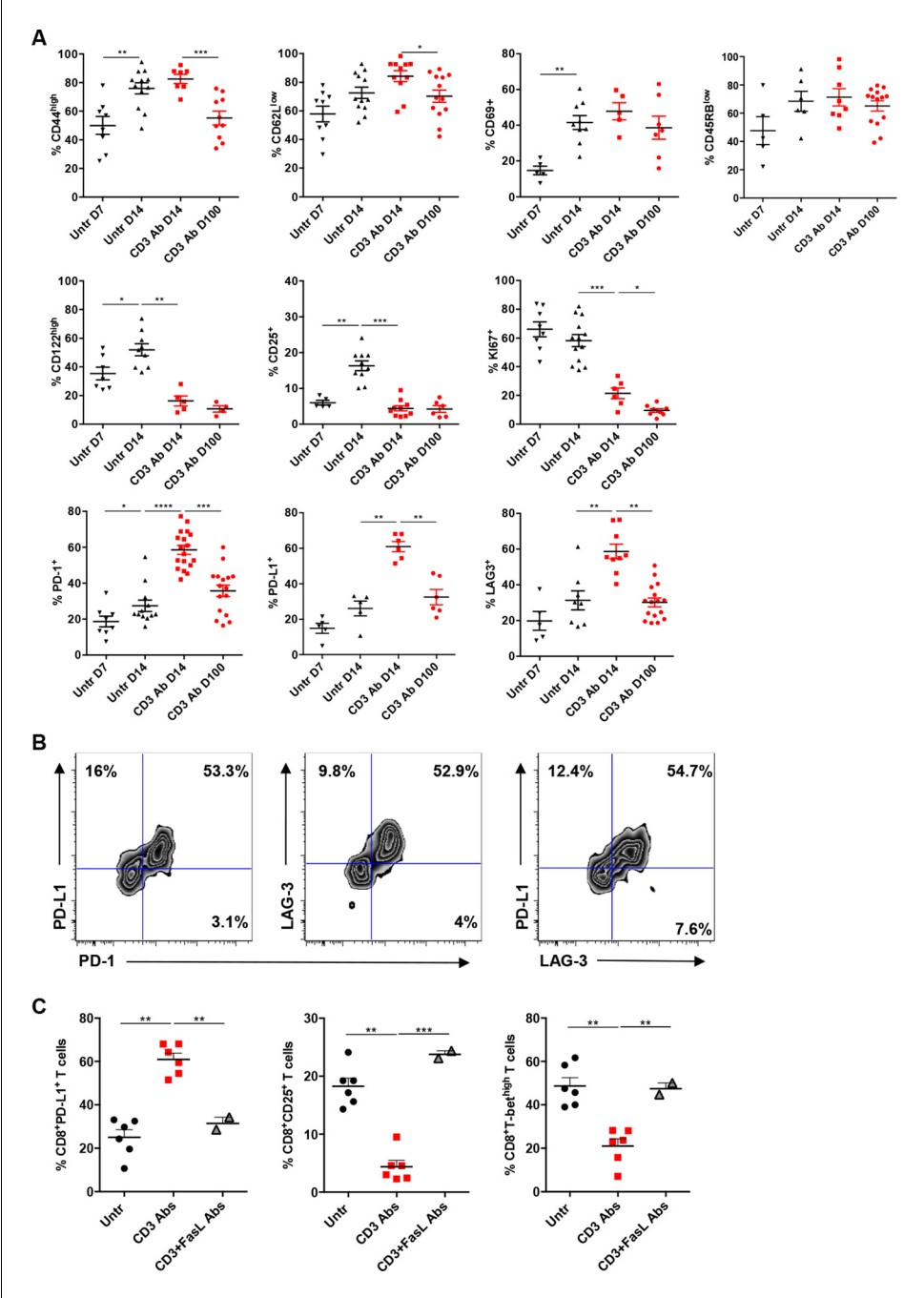

**Figure 3.** Phenotypic and functional characteristics of tolerant CD8[+] T cells. Pancreatic islet allografts were recovered from C57BL/6 mice after CD3 Ab treatment administered at day +7 post-transplant. (**A**) Expression of CD44, CD62, CD69, CD45RB, CD122, Ki67, CD25, PD-1, PD-L1 and LAG-3 (6–16/group) was evaluated on CD8[+] T cells on day +7, day +14 and day +100 post-transplant (*p<0.05, **p<0.01, ***p<0.001). (**B**) Co-expression of PD-1/PDL-1, PD-1/LAG-3 and PD-L1/LAG-3 on graft-infiltrating CD8[+] T cells recovered from CD3 Ab-treated mice on day +14 post-transplant. (**C**) Expression of PD-L1, CD25 and T-bet by graft infiltrating CD8[+] T cells isolated on day14 from untreated, CD3 Abs or CD3FasL Abs-treated recipients (n = 2–6/group) (**p<0.005).

The following figure supplements are available for figure 3:

**Figure supplement 1.** Mean fluorescence intensity of CD44 and CD62L expressed by intragraft CD8[+] T cells after CD3 Ab therapy.

**Figure supplement 2.** Phenotype of peripheral CD8[+] T cells after CD3 Ab therapy.

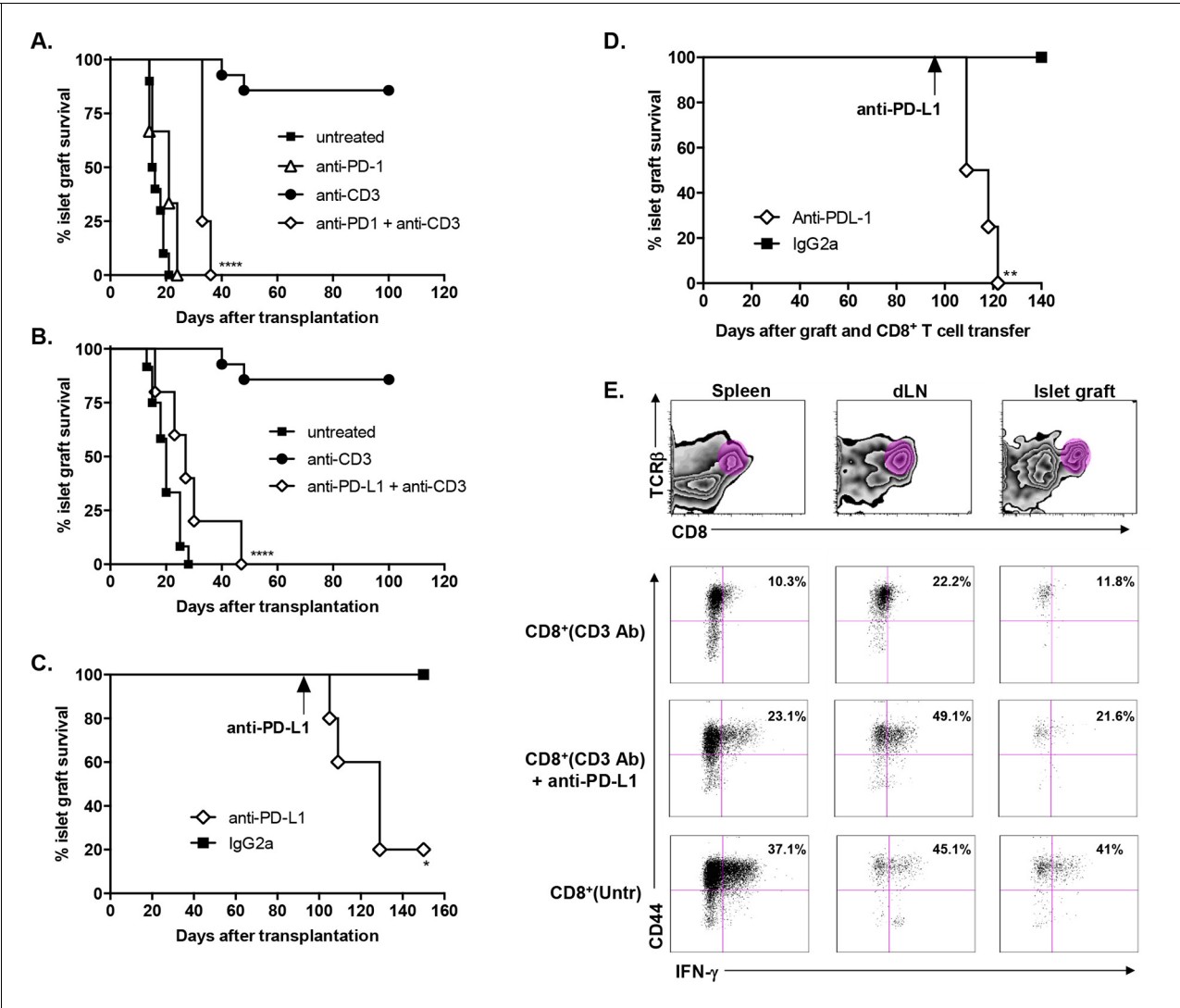

**Figure 4.** Transplant tolerance and CD8[+] T cell anergy rely on the PD-1/PDL-1 pathway. Graft survival of BALB/c islets was measured in C57BL/6 mice treated at day 7 with a combination of anti-CD3 F(ab')₂ and neutralizing antibodies against PD-1 (panel A, n = 5) or PD-L1 (panel B, n = 5) injected at the dose of 500 μg i.p. every other day for a total of 5 injections. (****p<0.0001 between anti-CD3 and anti-CD3+anti-PD-1/anti-PD-L1 Ab-treated mice). (C) C57BL/6 mice showing long-term islet graft acceptance after CD3 Ab therapy were treated on day +100 post-transplant with anti-PD-L1 antibodies or isotype control IgG2a (n = 5). Graft rejection occurred 2–3 weeks later (*p<0.03). (D) CD8[+] T lymphocytes were purified from the spleen of CD3 Ab-treated tolerant C57BL/6 mice and were transferred into C57BL/6 Rag[-/-] mice (3x10⁶/recipient). Recipient mice were grafted with pancreatic islets from BALB/c on day 0 and graft survival was monitored. On day +100 post-transplant, anti-PD-L1 antibodies or isotype control IgG2a were injected (n = 5) (**p<0.007). (E) Tolerant CD8[+] T cells were detected in the spleen, draining lymph nodes and islet allograft of C57BL/6 Rag[-/-] recipients. IFNγ production and CD44 expression were compared to the ones of CD8[+] T cells recovered after treatment with anti-PD-L1 antibodies or of CD8[+] T cells issued from untreated C57BL-6 mice and rejecting the islet graft.

The following figure supplement is available for figure 4:

**Figure supplement 1.** Administration of anti-PD-1 or anti-PD-L1 Abs reversed CD3 Ab-induced unresponsiveness of CD8[+] T cells.

Results showed that *Tgfb1* mRNA was expressed in 52% and 65% of CD4[+] and CD8[+] T cells, respectively (*Figure 6B*). In untreated recipients, only 9% of CD4[+] and 6% of CD8[+] intragraft T cells expressed *Tgfb1. Tgfb1* mRNA was expressed in very few T cells recovered from the spleen of the same untreated or CD3 Ab-treated mice. Administration of a neutralizing antibody to TGFβ partially, yet significantly, abrogated CD3 Ab-induced tolerance (*Figure 6C*). To assess whether TGFβ signaling within T cells was involved in the effect observed, we used as recipients transgenic DnTGFβRII

C57BL/6 mice that express a dominant negative form of the human TGFβ receptor II under the control of the mouse CD4 promoter (*Gorelik and Flavell, 2000*). In these mice, the dnTGFBRII-CD4 transgenic construct lacks the CD8 silencer. Thus, expression of the transgene blocks TGFβ signaling specifically in both CD4[+] and CD8[+] T cells. Treatment of these transgenic recipients with CD3 Ab on day +7 post-transplant did not induce long-term survival of islet allografts (median survival of 36.8 ± 6.1 days) (*Figure 6D*).

We asked whether TGFβ could influence the expression of PD-1 or PD-L1 on CD8[+] T cells. T cells were stimulated *in vitro* for 24 hr with CD3 Abs in presence of neutralizing Abs to TGFβ or recombinant TGFβ. Co-expression of PD-1 and PD-L1 was induced after TCR stimulation (*Figure 7—figure supplement 1*). Blockade of TGFβ downregulated expression of PD-1 and even more drastically that of PD-L1. In contrast, addition of recombinant TGFβ to the culture further promoted PD-1 but not PD-L1 expression. To assess the relevance of this pathway *in vivo*, islet allograft recipients were treated with CD3 Abs either alone or combined with TGFβ Abs. Paralleling the *in vitro* data, *in vivo* neutralization of TGFβ downregulated PD-1 and PD-L1 co-expression on graft infiltrating CD8[+] T cells (*Figure 7*).

## Gene signature of intragraft CD8[+] T cells

Using the Agilent platform, we compared intragraft CD8[+] T cells from CD3 Ab-treated tolerant mice to those from untreated mice showing acute rejection. Functional grouping analysis showed that gene categories related to effective T cell responses were downregulated in CD3 Ab treated mice as compared to controls (*Figure 8A*). Accordingly, pathways induced by IL-2 or TCR signaling were also downregulated (*Figure 8B* and *Figure 8—figure supplement 1*). Analysis of immune genes differentially expressed between the two groups revealed a decreased expression of genes involved in cell migration (*Cxcl9, Ccr10, Nrp1, Cd44*), proliferation (*Fos, Jun, Ccnd1*), co-stimulation (*Cd80, Icos, Cd200, Ctla4*), inflammatory cytokine signaling (*Tnf, Il2ra, Il1r2, Il18r1, Tnfrsf21*) and environmental sensing (hypoxia-inducible factor *Hif1a*) (*Figure 8C*).

In tolerant CD8[+] T cells, categories related to metabolic processes (nucleotides, energy, carbohydrates, lipids...) as well as cell cycle arrest were enriched. In addition, these cells showed upregulation of genes inhibiting or regulating T cell functions: inhibitors of cell proliferation (*Hopx, Cdkn2c, Map2k6, Mapk9*), of cytokine signaling (*Socs2, Socs6*), of differentiation (*Eid2, Id2*), of co-stimulation (*Btla*) (*Figure 8C* and *Figure 8—source data 1*). *Eomes*, encoding for the IFNγ transcription factor eomesodermin, was also upregulated in tolerant CD8 T cells.

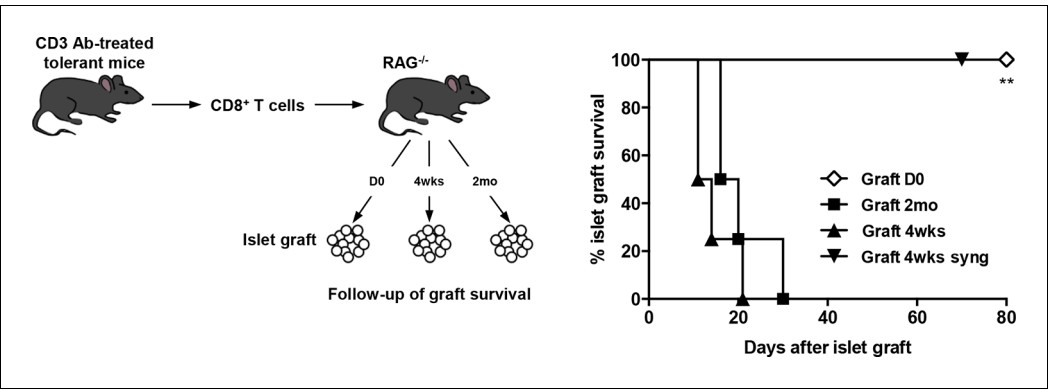

**Figure 5.** CD8[+] T cell anergy depends on the presence of the antigens. CD8[+] T lymphocytes were purified from CD3 Ab-treated tolerant C57BL/6 mice on day +100 post-transplant and were adoptively transferred into C57BL/6 Rag[-/-] mice (3x10[6]/recipient, day 0). C57BL/6 Rag[-/-] recipients were transplanted with pancreatic islets from BALB/c donors either on day 0 (D0, n = 4) or 4 weeks (4wks, n = 4) or 2 months (2mo, n = 4) after cell transfer (**p<0.007). Syngeneic islets from C57BL/6 donors were grafted 4 weeks after cell transfer and were used as controls.

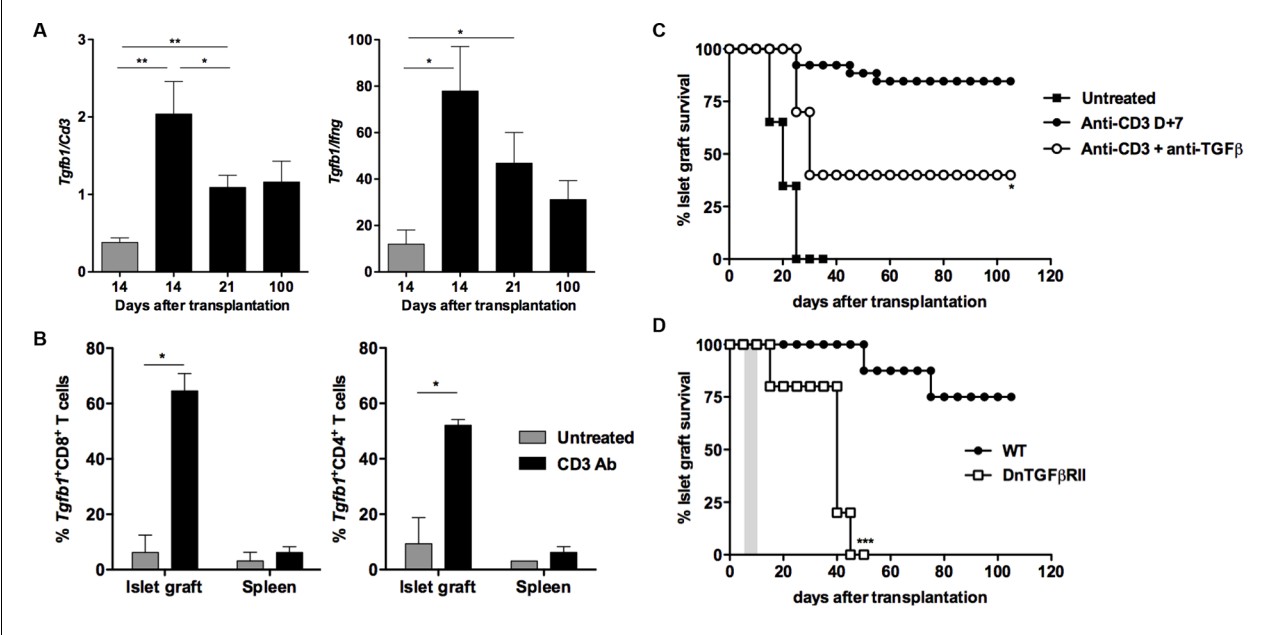

**Figure 6.** CD3 antibody-induced transplant tolerance depends on *in situ* TGFβ production and signaling in T cells. (**A**) Expression of *Tgfb1* mRNA and evaluation of the *Tgfb1/Ifng* ratio in pancreatic islet allografts recovered at day 14 from untreated mice or day 14, 21 and 100 after transplantation from CD3 Ab-treated recipients. Data are shown as mean ± SEM of 5–9 individual samples (*p<0.05, **p<0.01). (**B**) Single cell PCR: individual CD4[+] (n = 48) and CD8[+] (n = 60) T cells were sorted on day +14 post-transplant from the spleen or islet allografts recovered from C57BL/6 mice treated or not with CD3 antibodies. Expression of *Tgfb1* mRNA was measured in each cell. Results show the proportion of CD4[+] and CD8[+] T cells positive for *Tgfb* expression (*p<0.05). (**C**) Graft survival of BALB/c islets in wild-type C57BL/6 mice treated at day +7 with anti-CD3 F(ab')₂ and neutralizing TGFβ antibodies (1 mg/injection, twice a week for 3 weeks) (n = 4 to 8 per group, *p<0.05). (**D**) Abrogation of tolerance in DnTGFβRII C57BL/6 mice transplanted with BALB/c pancreatic islets and treated with CD3 antibodies on day7 post-transplant (n = 5 per group, ***p = 0.0002).

## Discussion

In the present study, using a pancreatic islet allograft model and CD3 Ab therapy, we provide novel insights into the immune mechanisms driving and sustaining CD8[+] T cell anergy thus leading to long-term graft survival. We demonstrate that CD3 Abs selectively deplete graft-infiltrating *Gzmb*[+]-*Prf1*[+] cytotoxic CD8[+] T cells by a FasL-mediated pathway. This depletion was accompanied by the onset of anergy in remaining CD8[+] T cells which was dependent on the presence of the alloantigens and on an *in situ* crosstalk between the PD-1/PD-L1 and TGFβ/TGFβRII pathways.

We previously reported that a short-term course with CD3 Abs induced long-term graft survival and antigen-specific tolerance provided the treatment was applied in a defined therapeutic window, that is at the time of effector T cells priming to the alloantigens (*Goto et al., 2013*; *You et al., 2012*). Mechanistic studies revealed that regulatory and effector T cells were differentially affected by the treatment. Foxp3 Tregs were relatively spared from CD3 Ab-induced depletion and could transfer antigen-specific tolerance, suggesting that they play an important role in sustaining graft survival (*Baas et al., 2013*; *Goto et al., 2013*; *You et al., 2012*). In contrast to Tregs, CD3 Abs preferentially induced apoptosis of activated effector T cells (*You et al., 2012*). Long-term graft survival correlated with an absence of donor-specific CD8[+] T cell responses at the periphery (*You et al., 2012*). Here, T cell unresponsiveness was confirmed at the tissue level as graft infiltrating T cells from CD3 Ab-treated recipient mice did not mount efficient IFNγ responses to donor antigens.

By single cell gene profiling using a method which proved optimal to discriminate functional patterns of CD8 T cells facing an immune stimulation (*Peixoto et al., 2007*; *Ribeiro-dos-Santos et al., 2012*), we identified two distinct subsets of graft infiltrating CD8[+] T cells, co-expressing either *Gzmb/Fasl* or *Gzmb/Prf1* mRNAs. Interestingly, *Gzmb*[+]*Prf1* cytotoxic CD8 T cells were selectively depleted after CD3 Ab therapy. The frequency of individual CD8 T cells expressing *Fasl* was enhanced at the end of CD3 Ab treatment but it was not associated with *Gzmb* coexpression in

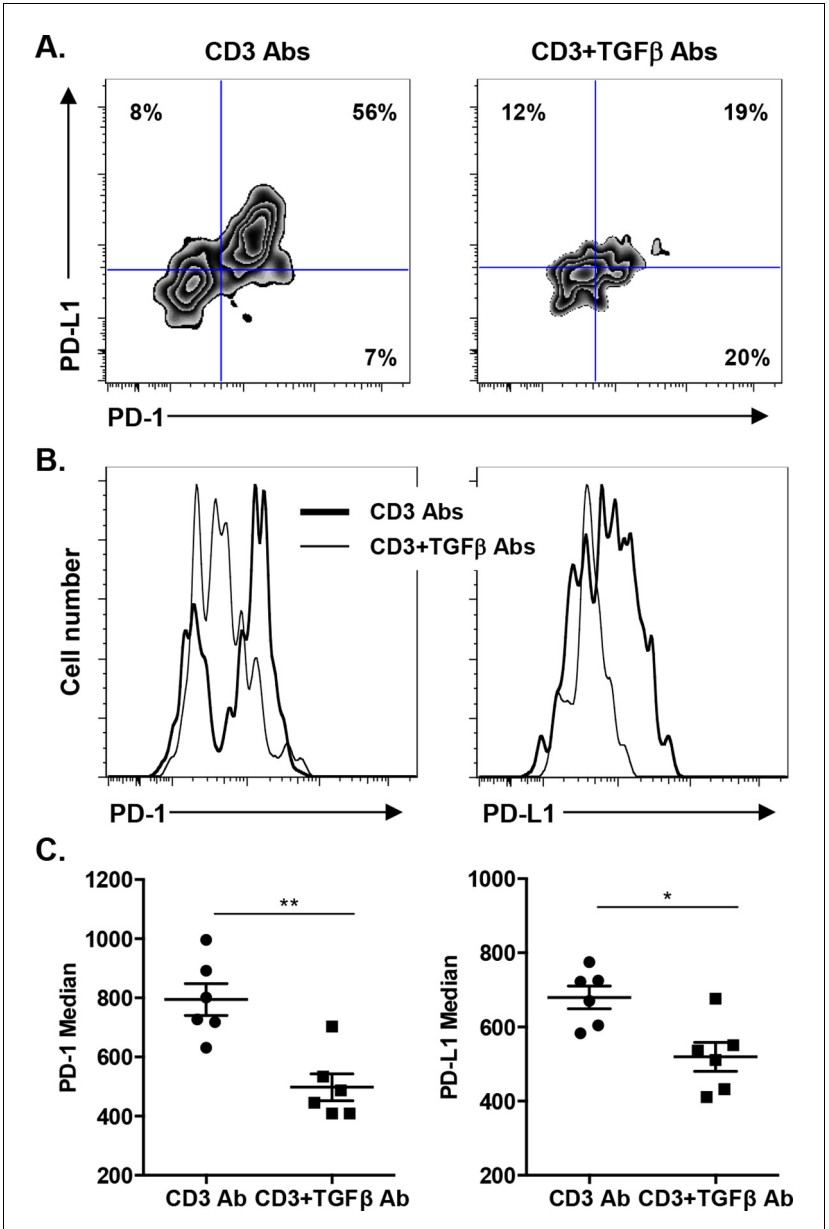

**Figure 7.** Induction of PD-1 and PD-L1 expression on intragraft CD8[+] T cells is regulated by TGFβ. C57BL/6 mice were transplanted with BALB/c pancreatic islets and treated at day 7 with anti-CD3 F(ab')$_2$ with or without neutralizing TGFβ antibodies. Mice were sacrificed on day 14 post-transplant and PD-1 and PD-L1 expressions were analyzed on graft-infiltrating CD8[+] T cells. (**A**) Co-expression of PD-1 and PD-L1 on CD8[+] T cells. (**B**) Histograms representing PD-1 and PD-L1 expression. (**C**) Median fluorescence intensity of PD-1 and PD-L1 (*p<0.02, **p<0.005).

The following figure supplement is available for figure 7:

**Figure supplement 1.** TGFβ modulates PD-1/PD-L1 expression on CD8[+] T cells.

contrast to what observed in untreated recipients. Flow cytometry analysis confirmed the increase of FasL[+]CD8[+] T cells. Previous studies showed that signaling through the TCR/CD3 complex promotes activation-induced cell death (AICD) in activated effector T cells through a Fas/FasL interaction, but independently of perforin and granzymes (*Brunner et al., 1995*; *Sobek et al., 2002*). T cells expressed both Fas and FasL after CD3 Ab treatment. Our results suggest, together with previous

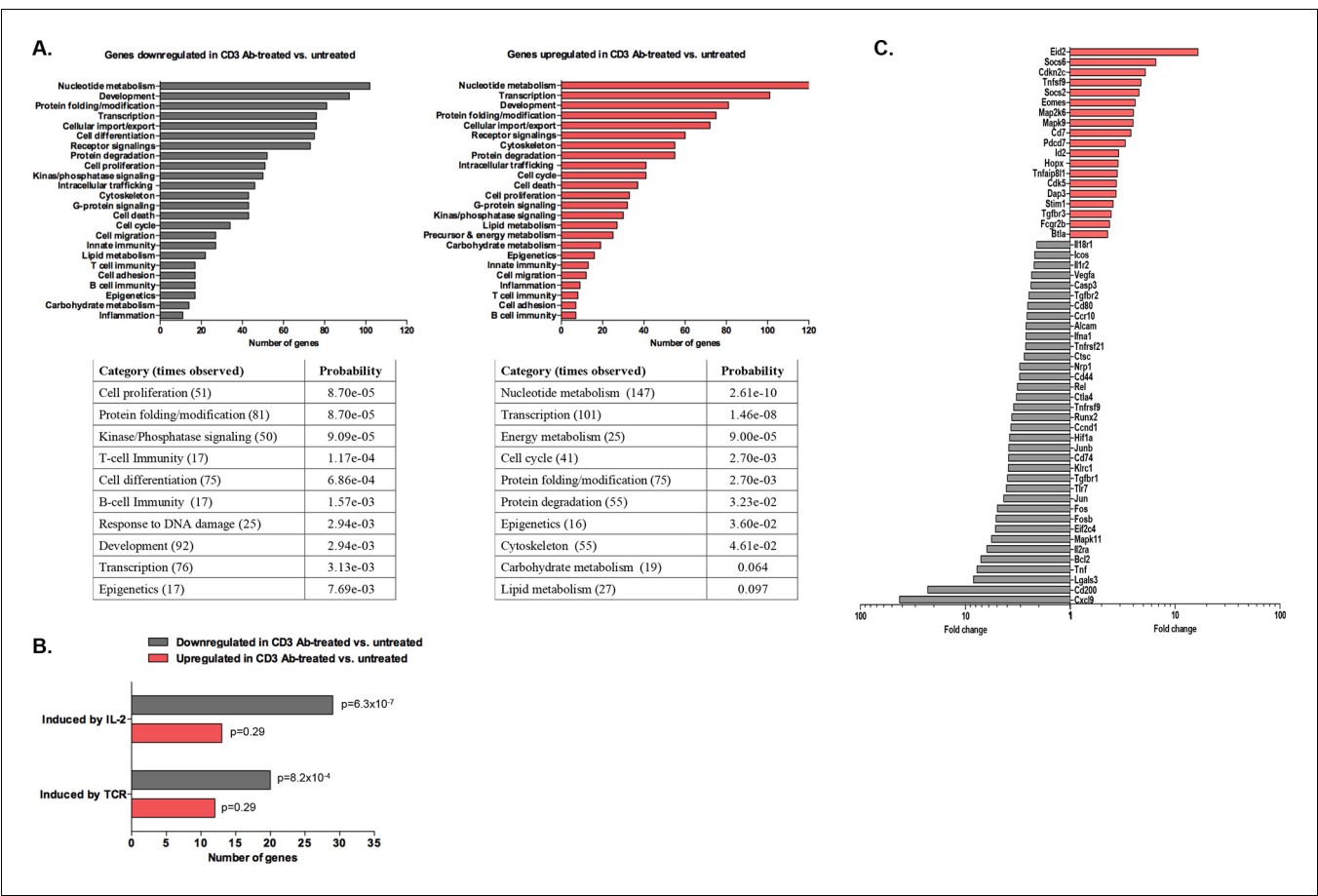

**Figure 8.** Transcriptomic analysis of tolerant intragraft CD8[+] T cells after CD3 Ab therapy. Five hundred graft-infiltrating CD8[+] T lymphocytes were sorted from untreated or CD3 Ab-treated C57BL/6 mice, on day +14 and day +100, respectively (n = 8 per group). Agilent Whole Mouse Genome Microarrays were performed after amplification of RNAs. Functional grouping analysis used annotations derived from Gene Ontology (fold-change >2, p<0.05). (**A**) Bar chart showing the frequency of representative categories downregulated (left panel) or upregulated (right panel) in tolerant CD8[+] T cells as compared to CD8[+] T cells from untreated recipient mice. Below, statistically enriched categories were indicated by their adjusted p-value (only the top 10 categories, Fisher's exact test with Benjamini-Hochberg correction for multiple testing). (**B**) Bar chart showing the frequency of target pathways induced by IL-2 ou TCR signaling. (**C**) Selection of immune genes that were downregulated (grey bars) or upregulated (red bars) in tolerant CD8[+] T cells as compared to CD8[+] T cells from untreated recipient mice (fold-change >2, p<0.05).

The following source data and figure supplement are available for figure 8:

**Source data 1.** Genes upregulated in anergic CD8[+] T cells from CD3 Ab-treated recipient mice.

**Figure supplement 1.** Heatmap of genes induced by IL-2 (**A**) or TCR (**B**) that are differentially expressed (>twofold, p<0.05) in CD8[+] T cells recovered from pancreatic islet allografts of CD3 antibody-treated tolerant mice (day +100 post-transplant) or of untreated mice (day +14 post-transplant).

reports using T cell hybridomas or clones, that apoptosis has occurred by triggering of Fas by FasL-expressing neighboring T cells (*Ayroldi et al., 1997*; *Brunner et al., 1995*). Rejection of islet allograft following FasL blockade further confirmed the Fas-dependence of CD3 antibody-induced depletion of alloreactive T cells. Therefore, our data concur to show that CD3 Abs induced apoptosis of *Gzmb*[+]*Prf1* CD8[+] T cells by a Fas/FasL-mediated pathway.

Apoptosis of alloreactive effectors was followed by the establishment of anergy in the remaining graft-infiltrating CD8[+] T cells. Single cell gene profiling showed that expression of *Gzmb*, *Prf1*, *Tbx21*, *Eomes* and *Klrg1* mRNAs decreased after CD3 Ab therapy as well as the simultaneous expression of 3 and more of the 7 genes tested. This downregulated gene pattern was even more obvious on day +100 post-transplant as intragraft CD8[+] T cells were characterized by a complete absence of co-expression of cytotoxic and inflammatory genes (i.e. 80% of the cells expressed no or

only one of the selected genes), a hallmark of profound intrinsic unresponsiveness. *Gzmb* was detected in 25% of the cells but was not associated with *Prf1* or *Fasl* showing that these CD8$^+$ T cells were deprived of effective killing ability.

Our findings are reminiscent of the pioneer experiments of Rocha and Von Boehmer who demonstrated the existence of anergy *in vivo* using TCR transgenic models. They described that female CD8 T cells specific for the male antigen HY rapidly expanded when transferred into male nude recipients and most of them died by AICD. The remaining transgenic CD8$^+$ T cells were intrinsically unresponsive to TCR stimuli (*Rocha and von Boehmer, 1991*). Importantly, when parked into a second female nude recipient for 2 months and subsequently transferred into a third male nude mice, the T cells regained their ability to respond to the HY antigen showing that antigen persistence was required to maintain CD8 T cell anergy (*Rocha et al., 1993*). In our model, the continuous presence of the alloantigen was essential to sustain CD8$^+$ T cell unresponsiveness *in vivo* as islet allografts transplanted 2 months after infusion of CD8$^+$ T cells from tolerant mice into RAG$^{-/-}$ recipients were rapidly rejected. Our data suggest that, within islet allografts, chronic antigen encounter induces a suboptimal activation that reinforces anergic signals in CD8$^+$ T cells and therefore sustains CD8 T cell tolerance.

Transcriptome analysis of graft infiltrating CD8$^+$ T cells recovered on day +100 post-transplant allowed us to identify a gene signature for tolerant T cells. Genes encoding for negative regulators of proliferation or differentiation (*Eid2, Cdkn2c, Hopx, Id2*) were overexpressed. *Hopx* (homeodomain-only protein), identified as a master regulator of the anergic state of induced Tregs, inhibits the expression of the AP-1 complex (*Hawiger et al., 2010*). Accordingly, *Fos* and *Jun* that compose the AP-1 complex were downregulated in tolerant CD8$^+$ T cells as well as their downstream target cyclin D1 (*Ccnd1*) (*Kang et al., 1992*; *Sundstedt et al., 1996*). Additionally, the decreased expression of *Cxcl9*, encoding an IFNγ inducible chemokine contributing to the recruitment of allograft reactive T cells (*Medoff et al., 2006*), as well as that of *Tnf, Il2ra, Lgals3, Il1r2, Il18r1* and *Tnfrsf9* revealed the inhibition of Th1 inflammatory responses. Tolerant CD8$^+$ T cells also downregulated the hypoxia-inducible factor *Hif1a* that has been shown to positively regulate T cell differentiation and effector functions by promoting aerobic glycolysis (*Doedens et al., 2013*; *Finlay et al., 2012*). Finally, we found an overexpression of *Eomes* in tolerant CD8$^+$ T cells. This was unexpected as eomesodermin (*Eomes*) plays well-described roles in cytotoxic CD8$^+$ T cell differentiation and memory formation (*Intlekofer et al., 2005*; *Pearce et al., 2003*). However, investigations in models of chronic infections in human and mouse (HIV, LCMV) have demonstrated that expression of Eomes by virus-specific CD8 T cells was associated with high expression of inhibitory receptors and impaired functions (*Buggert et al., 2014*; *Doering et al., 2012*; *Paley et al., 2012*). In our transplant model, as detailed below, the PD-1/PD-L1 pathway is mandatory for CD8 T cell tolerance. Thus, increased expression of Eomes in conjunction with PD-1/PD-L1 may characterize anergized CD8$^+$ T cells induced after CD3 Ab therapy. This finding highlights the issue of a molecular link between Eomes and PD-1/PD-L1. Lastly, one important feature common to our model and the infectious setting is the continuous presence of the cognate antigens. Therefore, we may hypothesize that chronic antigenic stimulation delivers signals inducing a preferential and sustained expression of Eomes.

Our study highlighted a key contribution of the PD-1/PD-L1 and the TGFβ/TGFβRII pathways and we identified a new role for TGFβ in modulating PD-1 and PD-L1 expression on CD8$^+$ T cells. After CD3 Ab therapy, graft-infiltrating CD8$^+$ T cells exhibited a CD44$^{high}$CD62L$^{low}$CD69$^+$CD45RB$^{low}$ antigen-experienced phenotype and coexpressed the inhibitory receptors PD-1, PDL-1 and LAG-3. The PD-1/PD-L1 pathway played a predominant role in the induction and the maintenance phase of CD3 Ab-mediated tolerance as neutralization of either receptor completely abrogated the therapeutic effect. More precisely, we demonstrated that the PD-1/PD-L1 signaling was mandatory for CD8$^+$ T cell unresponsiveness. Indeed, after treatment with CD3 antibodies, intragraft CD8$^+$ T cells showed a defective ability to proliferate, to produce IFNγ and to mount donor-specific responses. *In vivo* administration of anti-PD-1 or anti-PD-L1 antibodies restored the effector functions of allogeneic CD8$^+$ T cells, as shown by the up-regulation of the proliferation marker Ki67 and the IFNγ transcription factor T-bet as well as the regained ability to secrete IFNγ, and resulted in islet allograft rejection. Our findings are in accordance with the well-described role of the PD-1/PD-L1 pathway as a master regulator of immune responses notably through its ability to limit effector T cell interaction with dendritic cells and to abort their activation (*Fife et al., 2009*; *Francisco et al., 2010*). In various settings, PD-1 has been identified as a marker of dysfunctional CD8$^+$ T cells, and it contributed to

the establishment of tolerance (*Barber et al., 2006*; *Haspot et al., 2008*; *Ito et al., 2005*; *Lucas et al., 2011*; *Riella et al., 2012*).

We found that CD8$^+$ T cells coexpressed PD-1 and PD-L1, which may contribute to the formation of stable interactions with other CD8$^+$ T cells as well as other cells. PD-L1 is widely expressed on hematopoietic and nonhematopoietic cells and its expression increases with activation (*Keir et al., 2008*). In addition, aside from PD-1, PD-L1 can interact with B7-1 and a recent report demonstrated that this interaction contributed to the control of allogeneic T cell responses (*Keir et al., 2008*; *Yang et al., 2011*). We can thus hypothesize that PD-L1 engagement on graft-infiltrating CD8$^+$ T cells may deliver inhibitory signals and that these bidirectional interactions may promote and sustain T cell anergy and graft survival.

Another novel finding points to TGFβ as a regulator of PD-1 and PD-L1 expression and as a key mediator of CD3 Ab-induced allotolerance. We have previously shown in autoimmunity that CD3 antibodies cured type 1 diabetes and restored self-tolerance in a TGFβ-dependent manner (*Belghith et al., 2003*). The present results not only extend these findings to transplant tolerance but also provide new insights into the key regulatory role of this cytokine. First, the single cell PCR results showed that a large proportion of CD8$^+$ T cells (as well as CD4$^+$ T cells) present within the islet allografts produced TGFβ after CD3 antibody administration. This increased expression of TGFβ was restricted to the graft as it was not observed in the spleen. Secondly, we demonstrated that TGFβ signaling in T cells is mandatory for the induction of immune tolerance as CD3 Ab treatment failed to induce long-term islet graft survival in recipients presenting a T cell-selective mutated TGFβRII (*Gorelik and Flavell, 2000*). This result argues for an *in situ* autocrine/paracrine effect of TGFβ on allogeneic T cells. Third, we demonstrated a direct link between the TGFβ and the PD-1 pathways as TGFβ blockade, through the administration of neutralizing antibodies, downregulated PD-1 and PD-L1 expression on intragraft CD8$^+$ T cells and abrogated the tolerogenic properties of CD3 Abs. This inhibitory effect was even more drastic on PD-L1 than PD-1 showing that CD3 Ab-induced PD-L1 expression on CD8$^+$ T cells was highly dependent on TGFβ/TGFβR signaling.

In conclusion, our study highlighted new facets of immune mechanisms driving CD8$^+$ T cell peripheral tolerance and permanent acceptance of fully mismatched allografts after CD3 Ab therapy. Tolerance was established through elimination of highly cytotoxic CD8$^+$ T cells followed by the induction of CD8$^+$ T cells anergy which depended on a cross-talk between the PD-1/PD-L1 and TGFβ/TGFβRII pathways acting in an autocrine and paracrine manner in the graft environment. These mechanistic findings support the therapeutic potential of CD3 Ab which, concerning clinical translation, have shown potential value in the treatment of patients with new onset type 1 diabetes (*Herold et al., 2002*; *Keymeulen et al., 2005*; *Sherry et al., 2011*).

## Materials and methods

### Mice

C57BL/6, RAG$^{-/-}$ C57BL/6, and BALB/c female mice were bred in our facility under specific pathogen-free conditions. DnTGFβRII female C57BL/6 mice were obtained from the Jackson Laboratory (Bar Harbor, USA). Blood glucose was measured using ACCU-CHECK Performa glucometer (Roche Diagnostics, Meylan, France). Experiments were conducted according to European Directive (2010/63/UE) and were approved by the Ethical Committee of Paris Descartes University (registered number: 14–075).

### Pancreatic islet isolation and transplantation

Pancreatic islets were separated by density gradient centrifugation (Histopaque, Sigma-Aldrich, Lyon, France) after *in situ* digestion with collagenase P (Roche Diagnostics, Meylan, France) and transplanted (300 islets) under the kidney capsule of diabetic recipients. Diabetes was induced 3 to 4 days after a single injection of streptozotocin (Sigma-Aldrich) at 225 mg/kg. Diagnosis of graft rejection was made after three glucose measurements >250 mg/dl.

### Antibodies and *in vivo* treatments

The murine myeloma cell line SP2/0 producing the genetically engineered F(ab')$_2$ fragments of the hamster anti-mouse CD3ε antibody 145-2C11 (*Kostelny et al., 1992*) was provided by J.A.

Bluestone (UCSF, San Francisco, CA) The antibody was purified by protein G–Sepharose affinity chromatography. CD3 F(ab')$_2$ were injected i.v. at the dose of 50 µg/day for 5 consecutive days, starting on day 7 post-transplant. Anti-PD-L1 hybridoma (MIH5) was kindly provided by Pr. M. Azuma (Tokyo Medical and Dental University). Purified anti-PD-1 (RMP1-14) and anti-FasLigand antibodies (MLF4) were provided by Pr. H. Yagita (Juntendo University School of Medicine, Tokyo). TGFβ antibodies (2G.7) were produced in house. For flow cytometry, all antibodies were from BD Biosciences (Pharmingen, San Diego, CA, USA) except Foxp3 which was from eBioscience (San Diego, CA, USA).

## Adoptive transfers

RAG$^{-/-}$ C57BL/6 mice were reconstituted with 3.10$^6$ CD8$^+$ T cells isolated from the spleen of CD3 Ab-treated tolerant C57BL/6. The next day, recipients were transplanted with 300 BALB/c islets. PD-L1 antibodies were administered 100 days after transplantation. In another experiment, islet allografts were performed 4 weeks or 2 months after CD8$^+$ T cell transfer.

## RT-qPCR

mRNA was isolated using the mMACS isolation Kit and reverse transcribed into cDNA using the mMACS One-step cDNA Kit (MACS Molecular, Miltenyi Biotec, Bergisch Gladbach, Germany). RT-qPCR was performed on an ABI 7900HT fast real-time PCR system using primers, probes and master mixes from Applied Biosystems (Life Technologies, Carlsbad, CA, USA). HPRT was the housekeeping gene.

## Single cell PCR

Individual CD8$^+$ T cells were FACS sorted from the spleen or the islet allografts of untreated or CD3 Ab-treated recipients. After cell lysis by heating/cooling steps, RNA was specifically retrotranscribed using MuLV Reverse Transcriptase (Applied Biosystems) and 3' specific primers (Eurofins MWG, Ebersberg, Germany). The resulting cDNA was next amplified (first PCR with all primers). Product of this first PCR was then subjected to a second PCR using SYBR Green PCR Master Mix (Applied Biosystems) for each primers pairs. To ensure that each well contained a T cell, *Cd3e* mRNA was amplified simultaneously with the genes of interest. HPRT was the housekeeping gene. Multiplex single cell PCR was performed for the following genes: granzymes A and B (*Gzma, Gzmb*), perforin-1 (*Prf1*) and FasLigand (*Fasl*), which provide killing abilities, the transcription factors T-bet (*Tbx21*) and Eomesodermin (*Eomes*) which control IFNγ expression, and the killer-cell lectin like receptor G-1 (*Klrg1)* which marks terminally differentiated effector T cells.

## Microarray hybridization and analysis

For each sample, 500 graft-infiltrating CD8$^+$ T cells from individual tolerant CD3 Ab-treated recipients (n = 8) or untreated recipients (n = 8) were FACS sorted, lyzed using SuperAmp$^{TM}$lysis buffer (Miltenyi Biotec) and stored at -80°C until all samples were collected. Amplification of RNA, sample hydridization (Agilent whole mouse genome oligo microarrays), scanning and data acquisition were performed by Miltenyi Biotec. Functional grouping analysis has been performed using Gene Ontology. Gene expression data have been deposited in the GEO database (accession number GSE68208).

## Statistics

Cumulative graft survival was calculated using the Kaplan-Meier method. The statistical comparison was performed using the logrank (Mantel-Cox) test. When appropriate, the Student's t tests or *Chi* square ($\chi^2$) tests were used. A *p* value <0.05 was considered significant.

## Acknowledgements

The authors thank Claire Mangez for technical assistance and Jérome Megret for single cell sorting. We are also grateful to Pr. Azuma (Tokyo Medical and Dental University) for providing the anti-PD-L1 hybridoma.

## Additional information

### Funding

| Funder | Grant reference number | Author |
|---|---|---|
| European Commission | RISET, 512090 IP | Lucienne Chatenoud<br>Sylvaine You |
| Juvenile Diabetes Research Foundation | R11119KK | Lucienne Chatenoud<br>Sylvaine You |
| Institut National de la Santé et de la Recherche Médicale | Institutional funding | Lucienne Chatenoud<br>Sylvaine You |
| Université | Institutional funding | Lucienne Chatenoud<br>Sylvaine You |
| Nierstichting | KFB 10.003 | Marije Baas |

The funders had no role in study design, data collection and interpretation, or the decision to submit the work for publication.

### Author contributions

MB, SY, Conception and design, Acquisition of data, Analysis and interpretation of data, Drafting or revising the article; AB, Acquisition of data, Analysis and interpretation of data, Drafting or revising the article; TG, FV, EWE, Acquisition of data, Analysis and interpretation of data; HY, Provided purified anti-PD1 and anti-FasLigand antibodies in large amounts for in vivo experiments. These tools were essential to our research project. He provided advices on the dose and timing of injection. He also contributed to the discussion of the manuscript, Drafting or revising the article, Contributed unpublished essential data or reagents; BS, Acquisition of data, Analysis and interpretation of data, Contributed unpublished essential data or reagents; H-DV, Provided critical advices on the direction of some experiments and the discussion of the results. Conception and design, Analysis and interpretation of data; BR, Provided critical advices on the multiplex PCR performed on individual CD8+ T cells, from the conception of the experiments to the interpretation of the results. She also provided reagents for these single cell PCR., Conception and design, Analysis and interpretation of data, Contributed unpublished essential data or reagents; LC, Conception and design, Drafting or revising the article

### Ethics

Animal experimentation: This study was performed in strict accordance with the recommendations of the European Directive 2010/63/UE. All of the animals were handled according to approved institutional animal care and use committee protocols of the University of Paris Descartes.The protocol was approved by the Ethical Committee of Paris Descartes University (registered number: 14-075). All surgery was performed under ketamine/xylasine anesthesia, and every effort was made to minimize suffering.

## Additional files

### Major datasets

The following datasets were generated:

| Author(s) | Year | Dataset title | Dataset URL | Database, license, and accessibility information |
|---|---|---|---|---|
| You S | 2015 | Transcriptomic analysis of anergic CD8 T cells in transplant tolerance | http://www.ncbi.nlm.nih.gov/geo | Publicly available at the Gene Expression Omnibus (GSE68208) |

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
