## [Decision Letter]

Thank you for submitting your work entitled "Revisiting CD8^+^ T cell anergy in transplant tolerance: a new role for TGF-β in regulating the PD-1/PD-L1 pathway" for peer review at *eLife*. Your submission has been favorably evaluated by Tadatsugu Taniguchi (Senior editor) and four reviewers, one of whom is a member of our Board of Reviewing Editors.

The reviewers have discussed the reviews with one another and the Reviewing editor has drafted this decision to help you prepare a revised submission.

Summary:

The manuscript shows a very interesting set of data implicating a TGFβ-PD1 axis in generating and maintaining CD8 T cell allotolerance, in the context of non-FcR-binding anti-CD3 mAb-mediated induction of CD8 T cell allotolerance of solid islet transplants in vivo. It also indicates roles for Fas/FasL and loss of perforin-based lytic mechanisms in CD8 T cells in the success of the anti-CD3 therapy. The authors identify, by single-cell gene profiling, two distinct granzyme B-expressing CD8 T cell populations in grafts, one perforin-positive and the other FasL-positive. Their data suggest that the perforin-expressing cells are selectively depleted in anti-CD3 therapy in a Fas-dependent process, and the other population is rendered non-responsive over the long term by active repression via the PD1-PDL1 pathway in a process requiring continuous antigen exposure. The molecular mechanisms that account for these interactions have not been studied but would be of great interest.

Essential revisions:

Despite enthusiasm, major concerns remain to be addressed, with either textual modification/s and/or additional information and/or evidence where appropriate.

1) No rationale is provided for the selection of seven genes for the characterization of CD8 T cells, such as experimental evidence correlating their expression patterns with CD8 T cell functionality. Further, the data assume 'anergy' of the graft-infiltrating CD8 T cells (as well as CD8 T cells in lymphoid organs), yet show no assays demonstrating such anergy in cellular function. This is a particular concern since 'anergy' refers to the failure of the cells to reject allografts, yet cellular non-responsiveness appears to be through active inhibition because anti-PD1 or anti-PDL1 can reverse it. The term 'anergy' could use either replacement or extensive clarification.

2) The data provide no estimates of cell numbers anywhere, either in grafts from non-tolerant versus tolerant mice, or in the cell transfer experiments into RAG-null mice, making it difficult to assess the relative impact of cell depletion versus anergy. Cell populations need better definition and statistical assessment of their numbers and the changes therein over time of therapy.

3) The manuscript needs to acknowledge that the evidence for Fas-dependence of selective depletion of CD8 T cells expressing granzyme B and perforin is correlative as yet, and would require approaches such as FasL blockade to be definitive.

4) There seems an inconsistency between the amount of protein expression (Figure 1—figure supplement 2) and the frequency of protein-expressing cells (Figure 1). Thus, the amount of FasL normalized to CD3 is ~65% less in treated versus untreated mice at day 14 (Figure 1—figure supplement 2) while the frequency of FasL^+^ cells is ~50% MORE (Figure 1). The reduction of granzyme B is more drastic in amount (Figure 1—figure supplement 2) than in frequency (Figure 1). The criteria for the judgment of expression in Figure 1 should be carefully described, since if amounts of protein per cell are significantly different, the cells at different time points may have different origins or functions.

5) The data provide no evidence for the antigen-specificity of the tolerisation or of its maintenance in RAG-null mice, nor has the allospecificity of individual CD8 T cells been formally documented.

6) It is unclear from the description of the methods if the CD8 T cell transfers into RAG-null mice have used intragraft cells or splenic cells for the transfers. Since 3 million cells were transferred, it is likely that splenic T cells were used. The gene expression and phenotyping data show some differences between intragraft and splenic CD8 T cells from anti-CD3-tolerised mice. However, the intragraft CD8 T cells are extensively characterised, while splenic CD8 T cells allotolerise quite effectively. This makes it hard to interpret the mechanistic significance of the patterns observed in intragraft CD8 T cells.

[Editors' note: further revisions were requested prior to acceptance, as described below.]

Thank you for resubmitting your work entitled "TGFβ-dependent expression of PD-1 and PD-L1 controls CD8^+^ T cell anergy in transplant tolerance" for further consideration at *eLife*. Your revised article has been favorably evaluated by Tadatsugu Taniguchi (Senior editor), a Reviewing editor, and three reviewers. The manuscript has been improved but there are some remaining issues that need to be addressed before acceptance, as outlined below:

1) In Figure 2, the 'n' for the CD3+FasL abs appears to be 2. This is quite small. The related figure (2A) indicates that there were 4 animals studied. This needs to be clarified.

2) If there are data on the CD8^+^ cells from mice treated with the CD3+FasL abs, they would be useful to include.

3) CD3 antibody-induced upregulation of PD-L1 on intragraft CD8^+^ T cells was inhibited following in vivo FasL blockade; – it would be best to include Figure 3—figure supplement 3 in the main manuscript.

4) While Fas/FasL is a well-recognized pathway of cell death, it would be reasonable to address how this might be occurring – is it suicide or are neighboring cells providing the lethal ligand? Is Fas/FasL co-expression on the same cell functional? At the least, the issue needs to be acknowledged and discussed.

5) Clarification of Figure 4: Should there be a point at d100 when the anti-PD-L1 was given? In other words were the grafts monitored before d100 after anti-PD-L1 injection? Since the number of mice is small, is it certain that those that really reject after anti-PD-L1 really been non-rejecting before d100 when the Ab was given?

6) In the experiment in Figure 5, presumably tolerance was lost during the 2 months in the absence of the BALB/c islets. Did RAG^-/-^ recipients of spleen cells from the tolerant B6 mice not reject islets that were transplanted right after the transfer of cells? (Figure 5—figure supplement 1 does not address this control.)

7) Clarification of the implications of Figure 6: What explains the site differences in *Tgfb1* expression, since TGFb was induced in culture, and since the soluble cytokine would probably have effects at all sites?

8) The last sentence of Results – EOMES has functions other than serving as a transcription factor associated with Th1 cell development. It would be useful to discuss the possible explanation for its increase in the tolerized treated cells.

9) In the interests of brevity, the Discussion could be shortened in text that is not necessarily supported by the data, such as the extended PD-L1 expression discussion when PD-L1 expression was not studied in the islet grafts.

---

## [Author Response]

*Essential revisions:*

*Despite enthusiasm, major concerns remain to be addressed, with either textual modification/s and/or additional information and/or evidence where appropriate. 1) No rationale is provided for the selection of seven genes for the characterization of CD8 T cells, such as experimental evidence correlating their expression patterns with CD8 T cell functionality.*

We apologize for not having provided a clear rationale for the selection of the 7 genes analyzed. Indeed, these genes have been chosen for the following reasons:

First, in pancreatic islet allograft models, IFNγ is a key mediator of CD8^+^+ T cell-mediated graft destruction (Diamond AS et al. *J* Immunol 2000; Sleater M Am J Transplant 2007). Single disruption of either pathway does not prolong graft survival. In contrast, simultaneous loss of both pathways inhibited graft rejection.

Second, our unpublished results show that expression of *Gzmb, Prf1, Fasl, Ifnγ, Tbx21* and *Eomes* mRNAs increased with time within the islet allografts from day 7 to day 14 post-transplant in untreated mice (Figure 9).

Third, it has been shown by Dr. Benedita Rocha who set up the multiplex PCR on individual CD8^+^ T cells that the coexpression of effector genes, such as the ones we used, allows to identify CD8^+^ T cell subsets present at different phases of the immune responses and endowed with distinct inflammatory and/or killing ability (Peixoto A et al. Genome Research2004; Peixoto et al. J Exp Med 2007). We applied this methodology in the transplant setting to better dissect CD8^+^ T cell alloreactivity and behavior in response to CD3 antibody immunotherapy.

Therefore, we selected for our single cell PCR assays the transcription factors T-bet and Eomes which control IFNγ expression, granzymes A and B, perforin and FasLigand which provide killing capacities and KLRG1 which marks terminally differentiated effector T cells. These points and the corresponding references were added in the Results section (subsection “CD3 Ab therapy selectively depletes *Gzmb*^+^*Perf*^+^ CD8^+^ T cells and promotes anergy”) of the revised manuscript.

Author response image 1.Kinetics of intragraft effector gene expression in untreated recipients.Pancreatic islets from BALB/c mice were transplanted under the kidney capsule of C57BL/6 recipients. Allografts were recovered on day 7 and 14 post-transplant and expression *Gzmb, Prf1, Fasl Ifnγ, Tbx21* and *Eomes* mRNA was analyzed by RT-qPCR (n = 4-5) (*p<0.05).**DOI:**
http://dx.doi.org/10.7554/eLife.08133.020

*Further, the data assume 'anergy' of the graft-infiltrating CD8 T cells (as well as CD8 T cells in lymphoid organs), yet show no assays demonstrating such anergy in cellular function. This is a particular concern since 'anergy' refers to the failure of the cells to reject allografts, yet cellular non-responsiveness appears to be through active inhibition because anti-PD1 or anti-PDL1 can reverse it. The term 'anergy' could use either replacement or extensive clarification.*

We understand the editor and reviewers’ concern on the term ‘anergy’ that we used to define the CD8^+^ T cells recovered from tolerant mice after CD3 antibody therapy. We apologize as should have better recalled and discussed data obtained from our previous works where functional assays were performed. Indeed, both in a pancreatic islet and in a cardiac allograft models published in 2012 and 2013 (You S et al. Am J Transplant 2012; Goto R et al. Am J Transplant 2013), we showed, using a 20 hr IFNγ Elispot assay, that total spleen cells or purified CD8^+^ T cells from CD3 antibody-treated recipients were unable to respond when stimulated with donor antigens (response to third-party antigens being conserved). In addition, our unpublished results show that graft infiltrating T cells from CD3 Ab-treated recipients were unable to mount IFNγ responses (20 hr Elispot assay) towards BALB/c antigens as opposed to that recovered from untreated recipients. From these results, we concluded that T cells, and notably CD8^+^ T cells, from tolerant mice were unresponsive to their cognate antigens.

Results gained in the present manuscript from the single cell PCR, the transcriptome and the phenotype analysis, reinforced the ‘anergic’ nature of CD8^+^ T cells in tolerant mice.

Finally, our data further acknowledge the role of inhibitory receptors in the regulation of T cell anergy as suggested by several reports in different contexts (Shrikant P et al. Immunity 1999; Wells AD et al. J Clin Invest 2001; Greenwald RJ Immunity 2001; Tsushima F et al. Blood 2007; Fife BT et al. Immunol Rev 2008; Bishop KD et al. Cell Immunol 2009; Chikuma S et al. J Immunol 2009).

For the sake of clarification, we have now better described our previous data in the Introduction, Results(subsection “CD3 Ab therapy selectively depletes *Gzmb*^+^*Perf*^+^ CD8^+^ T cells and promotes anergy”) and Discussionsections and results from the IFNγ Elispot assay using graft infiltrating T cells have been included as new Figure 1—figure supplement 1.

*2) The data provide no estimates of cell numbers anywhere, either in grafts from non-tolerant versus tolerant mice, or in the cell transfer experiments into RAG-null mice, making it difficult to assess the relative impact of cell depletion versus anergy. Cell populations need better definition and statistical assessment of their numbers and the changes therein over time of therapy.*

T cell numbers and proportions in CD3 Ab treated recipient mice versus untreated mice were presented in details in our previous publications reporting the tolerogenic effect of CD3 antibodies in the transplant setting (You S et al. Am J Transplant 2012; Baas M et al. Transplant Proc 2013). We apologize for having not enough recalled these results in the present study, which is of importance for interpreting the data.

CD4^+^ and CD8^+^ T cells were transiently depleted in the grafts and the spleen after the 5-day treatment with CD3 antibodies. At the periphery T cells were back to their normal levels within 4-5 weeks post-treatment. As illustrated in Figure 10, in transplanted islets, numbers of CD8^+^ T cells remained low and stable up to day 100 after transplantation and treatment with CD3 antibodies (i.e. absence of significant de novoaccumulation after clearance of CD3 antibody from the blood circulation). Number of CD4^+^ T cells increased but remained significantly reduced over the long-term as compared to that detected in untreated recipients. This point has been clarified in the Introduction. In the RAG^-/-^ mice, no CD3 antibody treatment was applied. Only cell infusion was performed and islet graft survival was assessed.

Author response image 2.Intragraft T cell counts.Pancreatic islet allografts were recovered from untreated (left) or CD3 antibody-treated (right) mice and CD4^+^ and CD8^+^ T cells were counted at different time-points post-transplant.CD3 antibody treatment was applied on day +7 post-transplant for 5 days (gray area).**DOI:**
http://dx.doi.org/10.7554/eLife.08133.021

*3) The manuscript needs to acknowledge that the evidence for Fas-dependence of selective depletion of CD8 T cells expressing granzyme B and perforin is correlative as yet, and would require approaches such as FasL blockade to be definitive.*

As requested, we have now included results showing pancreatic islet allografts in mice treated with anti-FasL antibodies in conjunction with CD3 antibodies. These are presented in a new Figure 2. In these recipients, CD3 antibody therapy did not induce long-term graft survival and thereby immune tolerance. Administration of FasL antibodies abrogated CD3 antibody-induced T cell depletion, notably that of CD8^+^ T cells infiltrating the islet allografts (new Figure 2). As expected, FasL expression was inhibited on intragraft CD8^+^ T cells (new Figure 2). Interestingly, anti-FasL administration also inhibited the CD3 antibody-induced upregulation of the inhibitory receptor PD-L1 on intragraft CD8^+^ T cells (new Figure 3—figure supplement 3). These points are now included in the Resultssection, at the end of the subsection “CD3 Ab therapy selectively depletes *Gzmb*^+^*Perf*^+^ CD8^+^ T cells and promotes anergy” and in the subsection “Graft infiltrating CD8^+^ T cells present an inhibitory phenotype after CD3 Ab therapy” as well as in the third paragraph of the Discussion.

*4) There seems an inconsistency between the amount of protein expression (*Figure 1—figure supplement 2*) and the frequency of protein-expressing cells (*Figure 1*). Thus, the amount of FasL normalized to CD3 is ~65% less in treated versus untreated mice at day 14 (*Figure 1—figure supplement 2*) while the frequency of FasL^+^ cells is ~50% MORE (*Figure 1*). The reduction of granzyme B is more drastic in amount (*Figure 1—figure supplement 2*) than in frequency (*Figure 1*). The criteria for the judgment of expression in*
Figure 1
*should be carefully described, since if amounts of protein per cell are significantly different, the cells at different time points may have different origins or functions.*

We apologize for the apparent inconsistency. Figure 1 and Figure 1—figure supplement 2 show mRNA data obtained on distinct cell populations and using different methodologies which provide complementary results. Figure 1—figure supplement 2 represents PCR results on islet allografts as a whole and the analysis provided a global gene expression profile showing the downregulation of a cytotoxic/inflammatory signature after CD3 antibody therapy, including *Gzmb* and *Fasl*. Figure 1 focused on 72 individual intragraft CD8^+^ T cells subjected to multiplex PCR. As mRNAs were not quantified, we cannot exclude that although proportion of *Fasl* mRNA expressing CD8^+^ T cells increased after CD3 Ab treatment, the amount of *Fasl* mRNA per individual cells decreased. However, co-expression of inflammatory and cytotoxic genes provided important information on cell heterogeneity and on individual T cell behavior in the graft environment. Indeed, 2/3 of *Fasl* expressing CD8^+^ T cells from untreated recipients coexpressed *Gzmb*. In CD3 Ab treated hosts, only 1/3 coexpressed *Gzmb* and 1/3 did not coexpress any other inflammatory/cytotoxic genes. This suggests that *Fasl* mRNA expressing CD8^+^ T cells have distinct functional properties induced in response to the alloimmune stimulation and to CD3 antibody treatment.

The variable results between Figure 1 and Figure 1—figure supplement 2 further validate the benefit of performing single cell PCR experiments. Interestingly, flow cytometry analysis confirmed results obtained at the single cell level i.e. an increase of FasL protein expression on intragraft CD8^+^ T cells after CD3 antibody therapy (Figure 1). These points have now been discussed in the Discussion of the revised manuscript (third paragraph).

*5) The data provide no evidence for the antigen-specificity of the tolerisation or of its maintenance in RAG-null mice, nor has the allospecificity of individual CD8 T cells been formally documented.*

As for points #1 and 2, the antigenic specificity of the immune tolerance induced by CD3 antibodies towards pancreatic islet allografts have been demonstrated in our previous work (You S et al. Am J Transplant 2012). Recipient mice, that showed long-term islet graft survival after treatment with CD3 antibodies on day +7 after transplantation, were rendered diabetic again on day +100 post-transplant either by injecting a single high dose of streptozotocin or after the nephrectomy of the kidney bearing the islets grafts. All recipients showed hyperglycemia as assessed by blood glucose levels > 400 mg/dL. Second islet grafts from the original donors (BALB/c) or third party donors (C3H) were then performed on the controlateral kidney. BALB/c islets survived indefinitely in contrast to C3H islets that were rejected within 20-30 days, thereby demonstrating that antigen-specific tolerance had indeed been induced in CD3 antibody-treated recipients. This point has been added in the Introduction(fourth paragraph).

In addition, ex-vivoexperiments (IFNγ Elispot) showed that spleen CD8^+^ T cells from CD3 Ab-treated mice were unable to mount donor-specific responses while responses to third-party antigens were normal (You S et al. Am J Transplant 2012).

Finally, in contrast to BALB/c allogeneic pancreatic islets, syngeneic islets were not rejected in RAG^-/-^ recipients infused 4 weeks earlier with CD8^+^ T cells, highlighting the allospecificity of the CD8^+^ T cells recovered from CD3 Ab-treated mice. This new result is now illustrated on Figure 5—figure supplement 1.

*6) It is unclear from the description of the methods if the CD8 T cell transfers into RAG-null mice have used intragraft cells or splenic cells for the transfers. Since 3 million cells were transferred, it is likely that splenic T cells were used. The gene expression and phenotyping data show some differences between intragraft and splenic CD8 T cells from anti-CD3-tolerised mice. However, the intragraft CD8 T cells are extensively characterised, while splenic CD8 T cells allotolerise quite effectively. This makes it hard to interpret the mechanistic significance of the patterns observed in intragraft CD8 T cells.*

This point is well taken and we apologize for not having sufficiently detailed the design of the experiments. We used CD8_+_ T cells from the spleen of tolerant CD3 antibody-treated mice for cell transfer into RAG^-/-^ mice. This point has been clarified in the Results(subsection “Continuous presence of alloantigen and the PD-1/PD-L1 pathway are mandatory for the induction and maintenance of CD3 Ab-induced CD8^+^ T cell tolerance”) and Materials and methods(subsection “Adoptive transfers”). We used spleen cells for the following reasons. First, technically, it was not possible to recover CD8^+^ T cells from the islet allografts in sufficient number to perform transfer experiments. Secondly and more importantly, as mentioned in points #1 and 5, we showed in our previous work that spleen purified CD8^+^ T cells were unable to respond to donor antigens after CD3 antibody therapy (You S et al. Am J Transplant 2012). Thus, CD8^+^ T cell unresponsiveness was evidenced at the periphery, which reflected what observed within the islet allografts (i.e. an absence of graft destruction). Lastly, PD-1 and PD-L1 expressions were also upregulated on spleen CD8^+^ T cells after CD3 antibody therapy and up to day 100 post-transplant as compared to untreated recipients (Figure 3—figure supplement 2). Therefore, spleen CD8_+_ T cells exhibited key functional features that reflected what observed within the islet allografts and we decided to use them to investigate CD8^+^ T cell tolerance in transfer experiments.

[Editors' note: further revisions were requested prior to acceptance, as described below.]

*The manuscript has been improved but there are some remaining issues that need to be addressed before acceptance, as outlined below:*

*1) In Figure 2, the 'n' for the CD3+FasL abs appears to be 2. This is quite small. The related figure (2A) indicates that there were 4 animals studied. This needs to be clarified.*

We understand the reviewers’ concern. We have treated a total of 6 recipient mice with CD3 Abs combined to FasL Abs. Two of them were sacrificed on day 14 post- transplant, i.e. right after the last Ab injection, to evaluate the effect of FasL blockade on the survival and phenotype of intragraft CD8^+^ T cells (Figure 2 and new panel Figure 3). The remaining four were used to monitor graft survival or rejection illustrated on Figure 2. This point has been clarified in the legend of revised Figure 2 and of Figure 3.

*2) If there are data on the CD8^+^ cells from mice treated with the CD3+FasL abs, they would be useful to include.*

As requested, we have included new data on the phenotype of CD8_+_ T cells recovered from the islet allografts after the combined CD3+FasL Ab treatment. We now show that these CD8^+^ T cells express increased levels of CD25 and T-bet as compared to those issued from CD3 Ab alone-treated recipients, highlighting the recovery of effector functions after FasL blockade. These results are shown on revised Figure 3 (new panel) and are described in the subsection “Graft infiltrating CD8^+^ T cells present an inhibitory phenotype after CD3 Ab therapy.

*3) CD3 antibody-induced upregulation of PD-L1 on intragraft CD8^+^ T cells was inhibited following* in vivo *FasL blockade;* – *it would be best to include* Figure3—figure supplement 3 *in the main manuscript.*

As requested, we have included this result in the revised manuscript, Figure 3 (new panel).

*4) While Fas/FasL is a well-recognized pathway of cell death, it would be reasonable to address how this might be occurring* – *is it suicide or are neighboring cells providing the lethal ligand? Is Fas/FasL co-expression on the same cell functional? At the least, the issue needs to be acknowledged and discussed.*

As requested, we have discussed this issue in the third paragraph of the Discussion. We have demonstrated that T cells expressed both Fas and FasL after CD3 Ab treatment. Our results suggest, together with previous reports using T cell hybridomas or clones, that apoptosis has occurred by triggering of Fas by FasL-expressing neighboring T cells (Brunner T et al. Nature 1995, Ayroldi E et al. Blood 1997).

*5) Clarification of Figure 4: Should there be a point at d100 when the anti-PD-L1 was given? In other words were the grafts monitored before d100 after anti-PD-L1 injection? Since the number of mice is small, is it certain that those that really reject after anti-PD-L1 really been non-rejecting before d100 when the Ab was given?*

We apologize for not having provided a clear experimental design for Figure 4. Short course treatment (5 days) with CD3 antibodies induced permanent survival of BALB/c pancreatic islets transplanted under the kidney capsule of C57BL/6 mice. On day 100 post-transplant, i.e. once tolerance was established, recipient mice were treated with anti- PD-L1 antibodies or IgG2a isotype control. Graft rejection occurred 2-3 weeks later. For the sake of clarification, we have modified the text in the subsection “Continuous presence of alloantigen and triggering of the PD-1/PD-L1 pathway are mandatory for the induction and maintenance of CD3 Ab-induced CD8^+^ T cell tolerance” and the legend of Figure 4.

*6) In the experiment in Figure 5, presumably tolerance was lost during the 2 months in the absence of the BALB/c islets. Did RAG^-/-^ recipients of spleen cells from the tolerant B6 mice not reject islets that were transplanted right after the transfer of cells? (*Figure 5—figure supplement 1 *does not address this control.)*

As requested, we now show on one single figure (Figure 5) the graft survival rate obtained when islet transplantation was performed either on the day of CD8^+^ T cell transfer (D0) or 4 weeks (4wks) or 2 months after cell transfer (2mo). Survival of control syngeneic islet graft is also included on Figure 5. Therefore, Figure 5—figure supplement 1 has been removed. The text describing the results in the subsection “Continuous presence of alloantigen and triggering of the PD-1/PD-L1 pathway are mandatory for the induction and maintenance of CD3 Ab-induced CD8^+^ T cell tolerance”, has been modified.

7) Clarification of the implications of Figure 6: What explains the site differences in Tgfb1 expression, since TGFb was induced in culture, and since the soluble cytokine would probably have effects at all sites?

Figure 6 shows ex-vivo *Tgfb* mRNA expression by individual CD8^+^ and CD4^+^ T cells sorted from the spleen or the islet allografts of untreated or CD3 Ab-treated mice. Results showed that a large proportion of T cells present within the islet allografts, but not in the spleen, produced TGFβ after CD3 Ab administration. No culture was performed in these experiments. For the sake of clarity, we have modified the text in the subsection “Efficient TGFβ/TGFβR signaling promotes PD-1 and PD-L1 expression on graft infiltrating CD8^+^ T cells and is required for CD3 Ab-induced transplant tolerance.”

*8) The last sentence of Results* – *EOMES has functions other than serving as a transcription factor associated with Th1 cell development. It would be useful to discuss the possible explanation for its increase in the tolerized treated cells.*

The increased Eomes expression in CD3 Ab-induced tolerant CD8_+_ T cells is indeed one novel message of our work. Eomes was mostly shown to play key role in CD8^+^ T cell differentiation and acquisition of an effector memory phenotype (Intlekofer M et al. Nat Immunol 2005). However, investigations in models of chronic infections in human and mouse (HIV, LCMV) have demonstrated that expression of Eomes by virus-specific CD8^+^ T cells was associated with high expression of inhibitory receptors and impaired functions (Doering TA et al. Immunity 2012; Paley MA et al. Science 2012; Buggert M et al. Plos Pathogens 2014). In our transplant model, as detailed below, the PD-1/PD-L1 pathway is mandatory for CD8_+_ T cell tolerance. Thus, increased expression of Eomes in conjunction with PD-1/PD-L1 may characterize anergized CD8^+^ T cells induced after CD3 Ab therapy. This finding highlights the issue of a molecular link between Eomes and PD-1/PD-L1. Lastly, one important feature common to our model and the infectious setting is the continuous presence of the cognate antigens. Therefore, we may hypothesize that chronic antigenic stimulation delivers signals inducing a preferential and sustained expression of Eomes. This point has been included in the Discussion section, sixth paragraph.

*9) In the interests of brevity, the Discussion could be shortened in text that is not necessarily supported by the data, such as the extended PD-L1 expression discussion when PD-L1 expression was not studied in the islet grafts.*

As requested, the Discussion has been shortened to better focus the message of our work.